# Share or Not?
# Learning to Schedule Language-Specific Capacity for Multilingual Translation

**Biao Zhang**[1][*] **Ankur Bapna**[2], **Rico Sennrich**[3,1], **Orhan Firat**[2]
[1] School of Informatics, University of Edinburgh
[2] Google Research
[3] Department of Computational Linguistics, University of Zurich
`b.zhang@ed.ac.uk,ankurbpn@google.com,sennrich@cl.uzh.ch,orhanf@google.com`

## Abstract

Using a mix of shared and language-specific (LS) parameters has shown promise in multilingual neural machine translation (MNMT), but the question of when and where LS capacity matters most is still under-studied. We offer such a study by proposing conditional language-specific routing (CLSR). CLSR employs hard binary gates conditioned on token representations to dynamically select LS or shared paths. By manipulating these gates, it can schedule LS capacity across sub-layers in MNMT subject to the guidance of translation signals and budget constraints. Moreover, CLSR can easily scale up to massively multilingual settings. Experiments with Transformer on OPUS-100 and WMT datasets show that: 1) MNMT is sensitive to both the amount and the position of LS modeling: distributing 10%-30% LS computation to the top and/or bottom encoder/decoder layers delivers the best performance; and 2) one-to-many translation benefits more from CLSR compared to many-to-one translation, particularly with unbalanced training data. Our study further verifies the trade-off between the shared capacity and LS capacity for multilingual translation. We corroborate our analysis by confirming the soundness of our findings as foundation of our improved multilingual Transformers. Source code and models are available at https://github.com/bzhangGo/zero/tree/iclr2021_clsr.

## 1 Introduction

Model architecture design injects inductive biases to neural network layouts, allowing a learning algorithm to favor certain representations over others, independent of the observed data (Mitchell, 1980). In multilingual neural machine translation (MNMT), where the learning objective is commonly cast as a multi-task learning problem (Firat et al., 2016a; Ha et al., 2016; Johnson et al., 2017), the inductive bias researchers usually study is deciding on which components of the neural network to share between tasks (languages), and which components to leave specific to the task or language. These components can be entire layer stacks, individual layers or even some sub-layers (Sachan & Neubig, 2018; Blackwood et al., 2018; Wang et al., 2019; Zhu et al., 2020). Noticeably, the search space of which parameters to share and at which granularity grows rapidly, as we make neural networks large or increase the number of tasks (languages). This rapid expansion of the search space prevents us from exhaustively exploring the choice of sharing patterns in MNMT.

The incapability of full-space exploration motivates methods relying on heuristics (Sachan & Neubig, 2018), that lack flexibility when more languages are covered, or meta-learning (Platanios et al., 2018), that are often hard to scale. These limitations hinder their generalization to large-scale multilingual models, which is the very focus of our study. In large scale multilingual models, also known as massively multilingual models (Aharoni et al., 2019; Arivazhagan et al., 2019; Zhang et al., 2020b), hundreds of languages with varying amounts of training data, difficulty and linguistic properties are jointly trained together in a multi-task setup. While the joint training enables positive

---

[*]Work done while Biao Zhang was interning at Google Research.

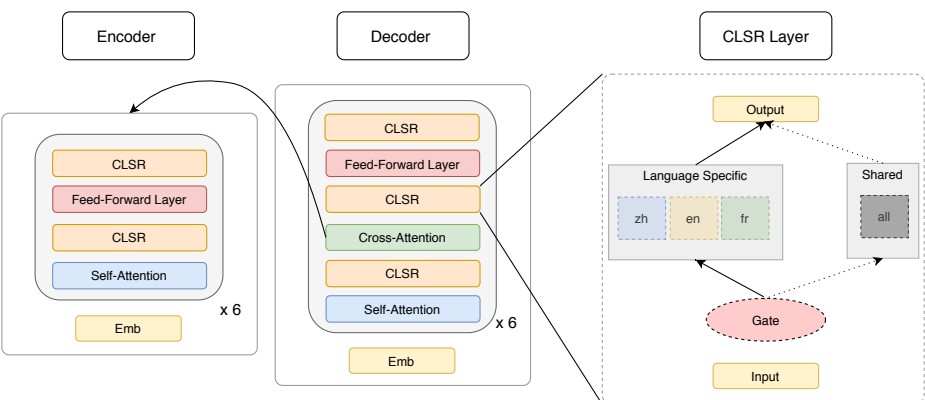

Figure 1: The model architecture used for our experiments. We introduce a CLSR layer after every transformer sub-layer in the encoder and the decoder. The gating layer learns to route every input through either the LS projection layer, or a shared projection layer. We analyze the outputs of the gating layers to develop a MNMT architecture with LS projections.

transfer across languages, it also introduces task-interference between dissimilar languages (Arivazhagan et al., 2019; Wang et al., 2020a;b) and a capacity bottleneck emerges due to the increased number of languages and data (Huang et al., 2019; Zhang et al., 2020b).

In this paper we adopt an end-to-end data driven approach (conditional language-specific routing, or CLSR) which permits directly probing a large section of the search space. We let the network learn the sharing structure from the data itself, by learning to route between language-specific (LS) or shared pathways. These two routes determine the mode of operation for the network: when the LS branch is selected, the model is given access to a set of LS layers (implemented as simple projections per language) and when the shared branch is chosen, the computation is routed to a layer that is used by all languages. By guiding the (gating) decision process with token level activation information, the network flexibly learns to alternate between the two modes and naturally lends itself to a conditional computation approach for multilingual processing (Bengio et al., 2013; Davis & Arel, 2013; Bapna et al., 2020). The gate states are optimized towards maximizing translation quality, but regularized with a budget constraint to control the amount of LS capacity[1]. Reducing the available budget results in fewer gates routing through the LS paths, enforcing CLSR to identify the most crucial sub-layers which allows us to observe and study the importance of each sub-layer for multilingual processing. Our approach is visually depicted in Figure 1.

We verify our proposal on WMT and the massively multilingual OPUS-100 dataset, with models building on the Transformer architecture (Vaswani et al., 2017). We explore target-specific and source-specific modeling for one-to-many[2] and many-to-one translation, respectively. To measure the degree of each sub-layer's tendency to be language-specific, we propose LSScore metric. Our results show that CLSR successfully navigates the trade-offs in LS modeling, outperforming several strong baselines. Our main findings are summarized below:

- Both the amount and the position of LS layers matter for MNMT. The best performance is achieved by distributing 10%-30% LS computation to the top and/or bottom encoder/decoder layers.
- Feed-forward sub-layers utilize more LS capacity compared to other sub-layers on one-to-many translation.
- One-to-many translation benefits more from CLSR (with target LS parameters) compared to many-to-one translation (with source LS parameters), particularly when the training data is imbalanced.
- The induced sharing pattern learned by CLSR is highly similar across languages.

---

[1]We use the term "*the amount of LS capacity*" to refer to the proportion of open gates where CLSR selects to route information through the LS path instead of its shared counterpart, which is directly regularized and guided by the budget constraint $p$ as in Eq. 6.

[2]In a one-to-many machine translation setup, a single source side language (commonly English) is tasked to be translated into multiple target languages, one at a time.

- We can use the learned patterns to hard-code better parameter sharing strategies for multi-lingual Transformers.

## 2 RELATED WORK

Our work closely relates to language-specific (LS) modeling for multilingual NMT and conditional computation for sequential data which we will recap both here. Early research on MNMT focused on improving shared capacity for separate bilingual models to enhance cross-lingual transfer. These efforts included sharing encoders for one-to-many translation (Dong et al., 2015), sharing decoders for many-to-one translation (Zoph & Knight, 2016; Lee et al., 2017) and sharing sub-layers (attention) for many-to-many translation (Firat et al., 2016a;b). These studies corroborated the feasibility of accommodating multiple languages with shared NMT sub-components, motivating researchers to explore universal MNMT. Ha et al. (2016) and Johnson et al. (2017) proposed such an implementation that performs multilingual translation with a single monolithic NMT model where the entire network is shared across languages, thanks to a target language token informing the model which language to translate into. Although this paradigm shows great scalability (Aharoni et al., 2019), the language token alone affords little flexibility in handling language diversity with a rigid share-all layout. Follow-up studies thus resort to LS modeling in an attempt to seek a better trade-off between sharing and not sharing. Methods in this category involve specializing neural attentions (Blackwood et al., 2018; Sachan & Neubig, 2018; Wang et al., 2019), broadening encoder outputs and normalizations (Zhang et al., 2020b), decoupling multilingual encoders and/or decoders (Vázquez et al., 2019; Escolano et al., 2020), using a fixed mix of LS and shared parameters (Wang et al., 2018), inserting lightweight adapters (Bapna & Firat, 2019) and separately modeling languages for different clusters (Tan et al., 2019), to name a few. Nevertheless, these methods heavily depend on heuristics, providing little evidence about how to optimally distribute LS capacity across the model.

By contrast, our proposed CLSR forces the model to learn LS behaviour. It can be treated as a simplified differentiable neural architecture search (NAS) model (Liu et al., 2019) with a search space defined by the presence/absence of LS projections after every transformer sub-layer. However, in contrast with NAS, we utilize conditional computation (Bengio et al., 2013) to make the choice of executing the shared or LS path conditional on the input representations. This allows us to compare and contrast the choice of LS vs shared paths on different inputs and languages. Conditional computation has previously been successfully applied to adapt the amount of computation to the input in recurrent models (Graves, 2016) and transformers (Dehghani et al., 2019; Elbayad et al., 2020), or to significantly scale up model capacity by utilizing sparsely-gated Mixture-of-Experts layers (Shazeer et al., 2017; Lepikhin et al., 2020). Zhang et al. (2020a) applied conditional computation to sparsify encoder outputs in sequence-to-sequence models in order to reduce attention costs, while Bapna et al. (2020) introduced conditional execution of Transformer sub-layers to control the amount of computation expended by the model at inference. Sukhbaatar et al. (2019) learn parameters that limit the attention spans, in order to make the attention operation more efficient, while Fan et al. (2020) utilize structured dropout to prune transformer layers at inference. Ruder et al. (2019) propose the sluice network that learns the inter-task (shared) sub-spaces on top of task-specific models. By contrast, CLSR starts with a totally shared model, and learns how to inject task-specific projections into it, which scales more easily to massively multilingual settings. Different from previous studies, we explore conditional computation as an analysis tool to understand the ideal arrangement of LS capacity for MNMT. Utilizing conditional computation to search for better LS sharing patterns in multilingual translation, to the best of our knowledge, has never been investigated before.

## 3 BACKGROUND: MNMT

Given a source sentence $X' = \{x_1, x_2, \ldots, x_I\}$ and its target translation $Y = \{y_1, y_2, \ldots, y_J\}$, we follow Johnson et al. (2017) to reuse standard bilingual NMT models for multilingual translation by altering the source input with a language token *lang*, i.e. changing $X'$ to $X = \{lang, x_1, x_2, \ldots, x_I\}$. Note that *lang* denotes the target language in one-to-many translation but source language in many-to-one translation.

We model translation from $X$ to $Y$ with Transformer (Vaswani et al., 2017). Transformer relies on the following residual-normalization structure (He et al., 2015; Ba et al., 2016) to smooth informa-

tion flow and avoid gradient vanishing and explosion:

$$\mathbf{z}^{l+1} = \text{LN}\left(\mathbf{z}^l + f\left(\mathbf{z}^l\right)\right), \tag{1}$$

where $l$ denotes layer depth and $\text{LN}(\cdot)$ is layer normalization (Ba et al., 2016). Function $f(\cdot)$ represents the basic building block in Transformer, such as attention network or feed-forward network. The encoder in Transformer is a stack of $L$ identical layers, with each layer involving a self-attention sub-layer (SAN) and a feed-forward sub-layer (FFN). The decoder uses a similar structure except for an extra cross-attention sub-layer (CAN) inserted in-between the above two sub-layers.

## 4 CONDITIONAL LANGUAGE-SPECIFIC ROUTING (CLSR)

The success of MNMT comes at the cost of expressivity and model's ability to capture language-specific characteristics. It has been empirically shown that the language signals from language indicator tokens alone are not sufficient (Arivazhagan et al., 2019), making architectures dedicated to LS modeling a necessity (Blackwood et al., 2018; Sachan & Neubig, 2018; Zhang et al., 2020b). Nevertheless, the question when and where LS modeling matters most in MNMT still remains to be answered. To this end, we propose conditional language-specific routing (CLSR) which specializes $f(\cdot)$ and changes the formulation in Equation 1 as follows:

$$\mathbf{z}^{l+1} = \text{LN}\left(\mathbf{z}^l + \text{CLSR}\left(f\left(\mathbf{z}^l\right)\right)\right). \tag{2}$$

CLSR learns a hard binary (scalar-valued) gate $g(\cdot)$ for each input token based on its hidden representation $\mathbf{z}^l \in \mathbb{R}^d$. These gates endow each sub-layer in Transformer with the capability of routing information selectively through either LS path $\mathbf{h}^{lang}$ or shared path $\mathbf{h}^{shared}$:

$$\text{CLSR}\left(f\left(\mathbf{z}^l\right)\right) = g(\mathbf{z}^l)\mathbf{h}^{lang} + (1 - g(\mathbf{z}^l))\mathbf{h}^{shared}, \tag{3}$$

$$\text{with} \quad \mathbf{h}^{lang} = f\left(\mathbf{z}^l\right)\mathbf{W}^{lang}, \quad \mathbf{h}^{shared} = f\left(\mathbf{z}^l\right)\mathbf{W}^{shared}, \tag{4}$$

where $\mathbf{W}^{shared}$ is a weight matrix shared across languages, while parameter $\mathbf{W}^{lang}$ is only used for modeling language *lang* which endows NMT with source or target LS modeling capacity.[3] Intuitively, a closed gate, corresponding to shared capacity, encourages maximal cross-lingual information transfer; an open gate, corresponding to LS capacity instead, improves language awareness for translation albeit it blocks knowledge transfer. CLSR balances between the two modes as controlled by the gates.

Following Bapna et al. (2020), we parameterize the gate $g(\cdot)$ with a two-layer feed-forward network $G(\cdot)$, and inject zero-mean Gaussian noise during training to discretize it (Chiu & Raffel, 2018) :

$$g(\mathbf{z}^l) = \sigma\left(G(\mathbf{z}^l) + \alpha(t)\mathcal{N}\left(0, 1\right)\right), \quad G(\mathbf{z}^l) = \text{Relu}\left(\mathbf{z}^l\mathbf{W}_1 + \mathbf{b}\right)\mathbf{w}_2, \tag{5}$$

where $\sigma(\cdot)$ is the logistic-sigmoid function, $d_g$ is the gating feed-forward hidden dimension, and $\mathbf{W}_1 \in \mathbb{R}^{d \times d_g}, \mathbf{w}_2 \in \mathbb{R}^{d_g}$ are trainable parameters. $\alpha$ is linearly increased along with training steps $t$. In this way, the gating parameters can be optimized with accurate gradients when $\alpha$ is small at the beginning so as to measure the degree to which each position in each sub-layer benefits from LS modeling. As training progresses, $\alpha$ grows larger, forcing the gating network to emit hard outputs. At inference time, we discretize the gate based on a simple decision rule: $g(\mathbf{z}^l) = \delta\left(G(\mathbf{z}^l) \geq 0\right)$, where $\delta(\cdot)$ is a Dirac measure.

We train the gates based on the standard maximum likelihood objective, along with an additional budget regularization term that enables control over the amount of LS capacity used for translation. Let the set of all CLSR layers in the encoder be $\mathcal{M}_{enc}$ and the decoder be $\mathcal{M}_{dec}$. Then the amount of LS computation utilized by a sentence pair $(X, Y)$ is given by $G_{(X,Y)} = \sum_{x \in X} \sum_{m \in \mathcal{M}_{enc}} g_m(x) + \sum_{y \in Y} \sum_{m \in \mathcal{M}_{dec}} g_m(y)$. Given a budget constraint $p \in [0, 1]$ and a batch of sentence pairs, $\mathcal{B}$, the training loss function of CLSR is formulated below:

$$\mathcal{L}\left(\mathcal{B}\right) = \sum_{(X,Y) \in \mathcal{B}} \text{MLE}\left(X, Y\right) + \left|\frac{\sum_{(X,Y) \in \mathcal{B}} G_{(X,Y)}}{\sum_{(X,Y) \in \mathcal{B}}(|X||\mathcal{M}_{enc}| + |Y||\mathcal{M}_{dec}|)} - p\right|, \tag{6}$$

---

[3]We maintain a set of language-specific weight matrices in order to compute $\mathbf{h}^{lang}$ for each language. To make the number of parameters manageable, we share the set of LS matrices $\mathbf{W}^{lang}$ across all encoder or decoder sub-layers, but distinguish $\mathbf{W}^{lang}_{enc}$ and $\mathbf{W}^{lang}_{dec}$, LS matrices for the encoder and decoder, respectively.

Intuitively, the budget constraint tries to regulate the amount of LS computation available to all tokens in the batch as a fraction $p$ of the total LS computation in the model. Given that we make a binary decision for every input for every CLSR layer, this corresponds to a search space of $\mathcal{O}(2^{|X||\mathcal{M}_{enc}|+|Y||\mathcal{M}_{dec}|})$. This gating space not only grows with respect to the model depth and sub-layer types, but also is highly input dependent. This dependency makes it difficult to search the entire space using heuristic methods (Blackwood et al., 2018).

The constraint in Equation 6 is imposed upon the aggregated gates. As a consequence, the model can learn to trade-off LS capacity for certain layers and inputs for others. Decreasing the budget encourages gate closure, such that the LS path is chosen only in the critical sub-layers. Thus, a properly learned gating function reveals the activations of LS paths and helps gain insights into the model behavior.

## 5 EXPERIMENTS

**Data and Evaluation**  We report results on two benchmarks: OPUS-100 (Zhang et al., 2020b) and WMT-14 (Barrault et al., 2019). OPUS-100 is a massively multilingual dataset collected from OPUS (Tiedemann, 2012), including 100 languages in total with 99 languages to-and-from English.[4] It consists of 55M training sentence pairs with up to 1M samples per language pair, and covers 94 dev/test language pairs, each with 2000 samples at most. WMT-14 is another multilingual English-centric dataset composed of 13 widely-used WMT benchmarks following Siddhant et al. (2020) but excluding Kazakh and Gujarati due to their poor parallel resource. Compared to OPUS-100, WMT-14 involves much fewer languages but its training data distribution is highly skewed across diverse language pairs, ranging from 0.2M (En-Tr) to 60M (En-Cs) training examples, thus posing severe challenges. We show more details about the train, dev and test data for WMT-14 in Appendix A.

We apply byte pair encoding (BPE) algorithm (Sennrich et al., 2016) using SentencePiece (Kudo & Richardson, 2018) to preprocess multilingual sentences with a vocabulary size of 64K. We use BLEU (Papineni et al., 2002), offered by SacreBLEU (Post, 2018)[5], for translation evaluation. Following Zhang et al. (2020b), we split the 94 test language pairs in OPUS-100 into three groups based on their training data size to ease model evaluation: high resource ($>$0.9M, 45 languages), low resource ($<$0.1M, 26 languages) and medium resource (others, 28 languages). Similarly, we split the 13 test language pairs in WMT-14 as follows: High ($>$10M, 5), Low ($<$1M, 5) and Med (others, 3).

We perform experiments for one-to-many translation (O2M) and many-to-one translation (M2O). In addition to using the original training data as is, we also report results with a temperature-based strategy to balance the training data distribution by over-sampling low-resource languages with a temperature of $T = 5$ (Arivazhagan et al., 2019).

We report average BLEU, and also show win ratio (Zhang et al., 2020b, WR), informing the proportion of language pairs on which our method beats our baseline. To evaluate how often each sub-layer is LS, we introduce a new metric, LSScore, formulated as follows:

$$\text{LSScore}_f(l,p) = \frac{1}{|\mathcal{P}|} \sum_{p \in \mathcal{P}} \tilde{g}^p_{l,f} - p, \tag{7}$$

where $\tilde{g}^p_{l,f}$ denotes the gating value averaged over all test tokens for the $l$-th sub-layer $f(\cdot)$ trained under a budget of $p$. Recall that we pose the budget constraint to the summed gate value instead of each individual gate, as in Equation 6. This gives CLSR the freedom to close more gates in some language-insensitive sub-layers (i.e. $\tilde{g}^p_{l,f} < p$) while preserve more for the others (i.e. $\tilde{g}^p_{l,f} > p$). A larger LSScore, $> 0$ in particular, indicates that this sub-layer utilizes more LS modeling.

**Model Settings**  We adapt Transformer base for our experiments: $L = 6$, $d = 512$, 8 attention heads with FFN middle size of 2048. Dropout of rate 0.1 is applied to residual connections and attention weights. We optimize parameters using Adam (Kingma & Ba, 2015) ($\beta_1 = 0.9, \beta_2 = 0.98$) with label smoothing of 0.1. Learning rate is scheduled according to the inverse square root of running steps with a warmup step of 4K (Vaswani et al., 2017). We limit training sequence length

---

[4]http://opus.nlpl.eu/opus-100.php
[5]Signature: BLEU+case.mixed+numrefs.1+smooth.exp+tok.13a+version.1.4.10.

to 100, and train all models with a batch size of 1920. We set the maximum running step to 500K and 600K for OPUS-100 and WMT-14, respectively. We perform beam search decoding with beam size of 4 and length penalty of 0.6. We average the last 5 checkpoints for evaluation. For CLSR, we set $d_g$ to 128, and linearly increase $\alpha$ from 0 to 5 along training steps (Bapna et al., 2020).[6] We vary the budget $p$ in the range of $\mathcal{P} = \{0.0, 0.1, 0.3, 0.5, 0.7, 0.9, 1.0\}$ to study its impact on model performance.

## 5.1 RESULTS ON OPUS-100

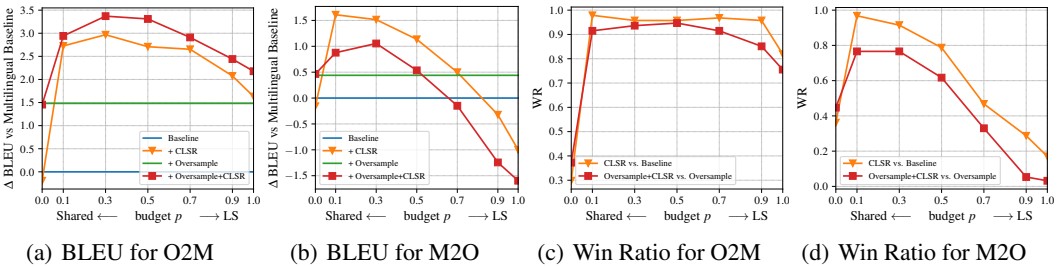

(a) BLEU for O2M      (b) BLEU for M2O      (c) Win Ratio for O2M      (d) Win Ratio for M2O

Figure 2: Average BLEU 2(a),2(b) and win ratio 2(c),2(d) over all test language pairs for O2M and M2O on OPUS-100 when varying the budget $p$. *Baseline*: multilingual baseline on the original training data; *Oversample*: oversampling low-resource data with a temperature of 5.

**On the trade-off between shared and language-specific capacity.** Using more LS modeling fails to deliver increasingly better translation performance, as shown in Figure 2(a) and 2(b) when $p$ approaches 1.0. At the point of full LS capacity for Transformer, i.e. $p = 1.0$, CLSR even underperforms its corresponding multilingual baseline by a large margin of 1.0-2.0 BLEU on M2O, Figure 2(b). Similarly, sharing all model parameters across language pairs, $p \rightarrow 0.0$, also yields suboptimal performance with either original or oversampled training data. CLSR achieves its best translation quality at $p = 0.3$ (+2.0/3.0 BLEU) and $p = 0.1$ (+0.5/1.5 BLEU) for O2M and M2O, respectively. The win ratio curves in Figure 2(c) and 2(d) further confirm the robustness of these quality improvements, where properly scheduling LS capacity outperforms the baselines on $> \sim 80\%$ language pairs. These results clearly show the trade-off between these two kinds of capacity.

**When does language-specific capacity matter for multilingual translation?** Results in Figure 2 show that O2M favors more LS modeling and benefits more from it compared to M2O ($p = 0.3$ vs. $p = 0.1$, and +3.0 vs. +1.5 BLEU on the original training data). We conjecture that translations on M2O share the same target language (English), so information can be easily transferred through shared modeling; by contrast, MNMT has to handle languages of diverse typological features for O2M, demanding stronger capacity delivered to each translation direction. Results in Figure 2 also show that CLSR yields more aggressive improvements on the original training data, compared to the oversampled counterpart (+3.0 vs. +2.0 BLEU on O2M and +1.5 vs. +0.5 BLEU on M2O). Fine-grained analysis on each resource group, as shown in Figure 7 (Appendix B), reveals that CLSR yields more improvements for low-resource translation where the oversampling strategy partially offsets these improvements.

**Where should we add language-specific capacity in multilingual Transformer?** Figure 2 suggests that CLSR uses 10-30% LS capacity to reach its best performance. We next study how CLSR schedules this capacity across all Transformer sub-layers, in order to determine the ideal arrangement of LS layers for translation. Figure 3 shows the results. Regardless of layer types, CLSR schedules more LS capacity to the top and/or bottom encoder/decoder layers rather than the middle ones. We find that more LS capacity is allocated to feed-forward sub-layers for O2M (both encoder Figure 3(a) and decoder Figure 3(b)), while the cross-attention sub-layers use more LS capacity for M2O (decoder, Figure 3(d)). Regarding the encoder for M2O, we observe no significant LSScore

---

[6]Note that, there are $99 \times 2$ and $13 \times 2$ independent $\mathbf{W}^{lang}$ weight matrices when CLSR is used for OPUS-100 and WMT-14, respectively. This corresponds to each language having a weight matrix in the encoder and the decoder.

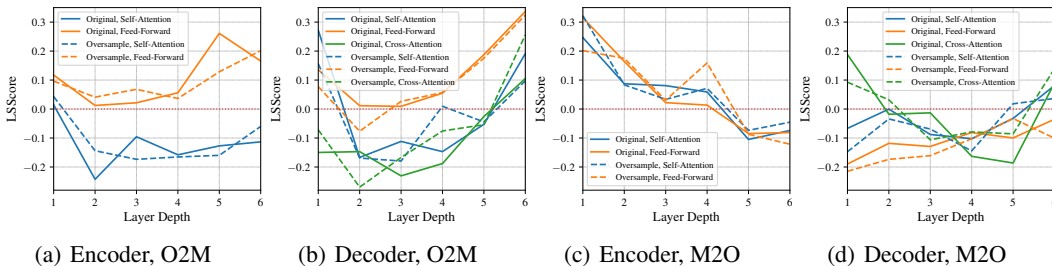

|  | (a) Encoder, O2M | (b) Decoder, O2M | (c) Encoder, M2O | (d) Decoder, M2O |

Figure 3: LSScore of encoder and decoder sub-layers for O2M 3(a), 3(b) and M2O 3(c), 3(d) on OPUS-100. The solid lines correspond to models trained on the original data, while the dashed lines are on the oversampled data. We also include a red, dotted line to indicate the LSScore of 0.

| Data Setting | Model | O2M | | | | | M2O | | | | |
|---|---|---|---|---|---|---|---|---|---|---|---|
| | | High | Med | Low | All | WR | High | Med | Low | All | WR |
| Original | Baseline | 21.39 | 22.36 | 18.02 | 20.93 | - | 28.55 | 30.10 | 29.71 | 29.27 | - |
| | LS$^\diamond$ | +0.75 | +1.83 | +3.96 | +1.79 | 94.68 | -0.54 | -0.16 | -0.46 | -0.41 | 32.98 |
| | CLSR-S | +0.06 | -0.09 | -0.34 | -0.08 | 41.49 | -0.14 | -0.06 | +0.02 | -0.08 | 37.23 |
| | CLSR-L | +0.39 | +1.54 | +4.37 | +1.62 | 84.04 | -1.13 | -0.47 | -0.46 | -0.78 | 21.28 |
| | CLSR$^\star$ | **+1.45** | **+2.83** | +6.40 | **+2.97** | 95.74 | **+0.65** | **+1.52** | **+3.79** | **+1.61** | **96.81** |
| | Top-Bottom | +1.27 | +2.71 | **+6.60** | +2.89 | 96.81 | +0.38 | +1.16 | +3.06 | +1.21 | 78.72 |
| | Dedicated | +1.35 | +2.75 | +6.46 | +2.90 | **97.87** | +0.61 | +1.50 | +2.85 | +1.38 | 89.36 |
| Over Sample | Baseline | 19.95 | 24.22 | 25.27 | 22.41 | - | 26.98 | 30.69 | **34.26** | 29.71 | - |
| | LS$^\diamond$ | +0.99 | +1.30 | +0.94 | +1.07 | 90.43 | -0.55 | -0.61 | -3.96 | -1.33 | 12.77 |
| | CLSR-S | +0.02 | +0.13 | +0.30 | +0.11 | 44.68 | -0.04 | +0.06 | -0.84 | -0.19 | 35.11 |
| | CLSR-L | +0.62 | +0.60 | +0.34 | +0.55 | 69.15 | -1.02 | -1.11 | -4.56 | -1.84 | 7.45 |
| | CLSR$^\star$ | +1.76 | **+2.04** | +1.94 | +1.89 | 93.62 | +0.82 | +0.84 | -0.13 | **+0.62** | 76.60 |
| | Top-Bottom | +1.73 | +1.91 | +1.83 | +1.81 | **96.81** | +0.83 | **+1.05** | -1.36 | +0.41 | 73.40 |
| | Dedicated | **+1.79** | +2.03 | **+2.07** | **+1.92** | 94.68 | **+0.99** | +0.92 | -0.88 | +0.55 | **77.66** |

Table 1: Translation quality for O2M and M2O on OPUS-100 with the original and oversampled training data. We list average BLEU↑ for High, Med, Low and All language groups, as well as WR↑ over all language pairs. *Baseline*: the vanilla multilingual baseline; *LS$^\diamond$*: the LS model proposed by Zhang et al. (2020b); *CLSR-S*: CLSR but always using shared modeling, i.e. $g(\mathbf{z}^l) = 0$ for all inputs; *CLSR-L*: CLSR but always using LS modeling, i.e. $g(\mathbf{z}^l) = 1$ for all inputs; *CLSR$^\star$*: the best CLSR model; *Top-Bottom*: applying LS modeling only to the top and bottom Transformer layers; *Dedicated*: dedicated model that allocates LS modeling based on LSScore distribution. Best results are highlighted in **bold**.

difference among different sub-layers as in Figure 3(c). Overall, CLSR tends to make the "M" side more LS, i.e. the encoder side of M2O (Figure 3(c)) and the decoder side of O2M (Figure 3(b)).

**Does CLSR schedule language-specific capacity based on linguistic similarity?** It is intriguing to explore how the capacity scheduled by CLSR is actualized, especially whether CLSR learns to organize LS capacity according to linguistic characteristics or not. The heatmap in Figure 6, Appendix B shows that the LSScore distribution over different sub-layers (y-axis) has only subtle difference across different language pairs (x-axis). This suggests that the capacity schedule has little to do with linguistic characteristics. More results in other settings are given in Figure 6, Appendix B, which reflect similar observation. In short, CLSR allocates LS capacity to specific sub-layers rather than specific languages. This could be ascribed to the design of CLSR. CLSR shares the gating parameters and the budget, $p$, across all languages, which might impose some inductive bias discouraging its LS behavior. Besides, the structure of the gating in CLSR (Eq. 5) might fail to offer enough flexibility for controlling the gates in different ways across languages and layers. We leave further study of LS gating to future work.

**On detailed results and comparison to other baselines.** Table 1 summarizes our results.[7] Although LS$^\diamond$ (Zhang et al., 2020b) improves O2M, it fails to surpass the vanilla Baseline on M2O, indicating that the position of its LS layer, i.e. on top of the encoder outputs, is sub-optimal for

---

[7]We list the number of trainable parameters for each model in Table 6, Appendix D.

many-to-one translation. Compared to LS$^\diamond$, CLSR-S uses no LS modeling while CLSR-L injects LS projection into each sub-layer, both of which delivers inferior performance on O2M (against LS$^\diamond$ and Baseline) and M2O (against Baseline), echoing our findings from Figure 2. By contrast, CLSR$^\star$, which uses an optimized budget p, yields promising results, beating Baseline on O2M and M2O, underlining the importance of seeking balances between sharing and not sharing.

**Can we use these insights to improve multilingual Transformer?** We answer this question by transferring our findings from Figure 3 into the following two Transformer variants: one enhances the top and bottom encoder/decoder layers with LS projections (*Top-Bottom*), while the other makes high-LSScore sub-layers LS (*Dedicated*).[8] Results of these experiments, in Table 1, demonstrate that incorporating our findings into Transformer helps to improve translation performance. The dedicated model in particular recovers and even surpasses the performance of CLSR$^\star$.

## 5.2 RESULTS ON WMT-14

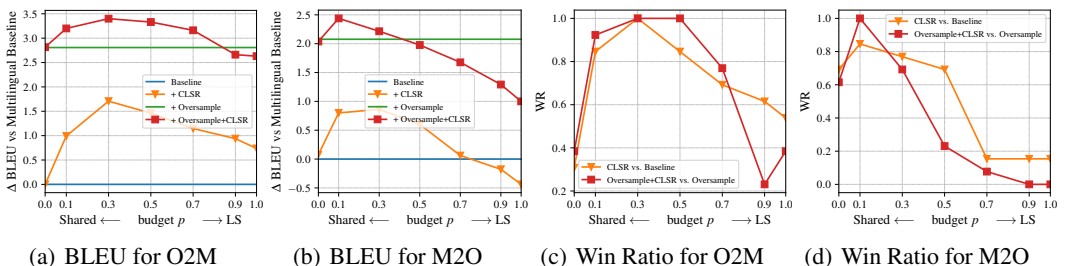

(a) BLEU for O2M  (b) BLEU for M2O  (c) Win Ratio for O2M  (d) Win Ratio for M2O

Figure 4: Average BLEU 4(a),4(b) and win ratio 4(c),4(d) over all test language pairs for O2M and M2O on WMT-14 when varying the budget $p$.

| Data Setting | Model | O2M | | | | | M2O | | | | |
|---|---|---|---|---|---|---|---|---|---|---|---|
| | | High | Med | Low | All | WR | High | Med | Low | All | WR |
| Original | Baseline | 27.24 | 16.27 | 8.72 | 17.58 | - | 30.64 | 24.37 | 18.66 | 24.58 | - |
| | LS$^\diamond$ | +0.06 | +0.66 | +1.68 | +0.83 | 84.62 | -0.06 | -0.37 | -0.92 | -0.46 | 23.08 |
| | CLSR-S | +0.00 | +0.16 | +0.20 | +0.12 | 53.85 | +0.04 | +0.16 | +0.28 | +0.17 | 69.23 |
| | CLSR-L | -0.52 | +0.43 | +2.06 | +0.70 | 61.54 | -0.82 | -0.50 | +0.00 | -0.43 | 15.38 |
| | CLSR$^\star$ | **+0.46** | **+1.46** | **+3.10** | **+1.71** | **100.0** | **+0.12** | +0.73 | +1.52 | +0.80 | **84.62** |
| | Top-Bottom | +0.10 | +0.96 | +2.78 | +1.34 | 84.62 | -0.08 | +0.70 | +1.26 | +0.62 | 61.54 |
| | Dedicated | +0.26 | +1.00 | +2.98 | +1.48 | 92.31 | +0.04 | **+0.86** | **+1.90** | **+0.95** | 84.62 |
| Over Sample | Baseline | 26.10 | 19.00 | 15.52 | 20.39 | - | 29.96 | 26.33 | 23.56 | 26.66 | - |
| | LS$^\diamond$ | +0.30 | +0.60 | +0.26 | +0.36 | 84.62 | -0.44 | -0.56 | -0.80 | -0.61 | 00.00 |
| | CLSR-S | -0.12 | +0.03 | +0.04 | -0.02 | 46.15 | +0.06 | +0.04 | -0.26 | -0.07 | 46.15 |
| | CLSR-L | -0.56 | -0.10 | -0.32 | -0.36 | 7.69 | -0.90 | -1.00 | -1.24 | -1.05 | 00.00 |
| | CLSR$^\star$ | **+0.50** | **+0.67** | +0.64 | **+0.59** | **100.0** | **+0.26** | **+0.30** | +0.50 | **+0.36** | **100.0** |
| | Top-Bottom | +0.12 | +0.40 | +0.56 | +0.36 | 84.62 | -0.08 | +0.00 | -0.50 | -0.22 | 30.77 |
| | Dedicated | +0.46 | +0.57 | **+0.66** | +0.56 | **100.0** | +0.16 | +0.17 | +0.22 | +0.19 | 84.62 |

Table 2: Translation quality for O2M and M2O on WMT-14 with the original and oversampled training data.

Figure 4 shows the capacity trade-off on WMT-14, reconfirming the ability of CLSR. One noticeable difference is that the relative improvements become smaller. We ascribe this to the smaller number of language pairs in WMT-14, where the effect of introducing LS capacity into the model is smaller compared to the massively multilingual setting (Arivazhagan et al., 2019). Table 2 shows similar results as Table 1, where CLSR$^\star$ outperforms both fully-shared (CLSR-S) and fully-LS (CLSR-L) baselines, and both Top-Bottom and Dedicated improve multilingual translation compared to Baseline (except for M2O with oversampling). More results are given in Appendix C. Our results demonstrate that the CLSR approach generalizes to different datasets, with varying number of lan-

---

[8]Detail about which sub-layers are specialized in Dedicated is given in Table 4, Appendix B.

guages and resource sizes. We provide additional experiments ablating different elements on CLSR in Table 7, Appendix E.

## 6 CONCLUSION AND FUTURE WORK

Share or not? This is an open question when developing MNMT models. In this paper, we attempt to answer this question by proposing conditional language-specific routing (CLSR). Our empirical results demonstrate that CLSR learns to balance between shared and LS paths across all NMT sub-layers, improving the quality of multilingual translation. Our analysis on OPUS-100 and WMT-14 suggest that both the position and the amount of LS capacity greatly affects MNMT. Scheduling 10%-30% LS layers to the top and/or bottom encoder/decoder layers reaches CLSR's peak performance. We also demonstrate how our findings can be leveraged to design a multilingual Transformer with an optimal sharing pattern. We believe that our work improves our understanding on the trade-off between sharing and not sharing, paving the way for better multilingual models.

In the future, we plan to extend our study to many-to-many translation as well as larger-capacity models. We also plan to adapt CLSR to other multilingual multi-task learning settings to better handle knowledge transfer among different tasks, especially for cross-lingual downstream transfer.

## ACKNOWLEDGEMENTS

We would like to thank Yuan Cao for his valuable feedback. We would also like to thank the Google Translate team for their constructive discussions and comments. We thank the reviewers for their insightful comments. Rico Sennrich has received funding from the Swiss National Science Foundation (project no. 176727).

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

# A    DATASET DETAILS

Table 3: Statistics of train, dev and test data for WMT-14.

| Language Pair | Data Sources | | | # Samples | | |
|---|---|---|---|---|---|---|
| | Train | Dev | Test | Train | Dev | Test |
| En-Cs | WMT19 | WMT17 | WMT18 | 64336053 | 3005 | 2983 |
| En-Fr | WMT15 | WMT13 | WMT14 | 40449146 | 3000 | 3003 |
| En-Ru | WMT19 | WMT18 | WMT19 | 38492126 | 3000 | 2000 |
| En-Zh | WMT19 | WMT18 | WMT19 | 25986436 | 3981 | 2000 |
| En-Es | WMT13 | WMT13 | WMT13 | 15182374 | 3004 | 3000 |
| En-Fi | WMT19 | WMT18 | WMT19 | 6587448 | 3000 | 1996 |
| En-De | WMT14 | WMT13 | WMT14 | 4508785 | 3000 | 3003 |
| En-Et | WMT18 | WMT18 | WMT18 | 2175873 | 2000 | 2000 |
| En-Lv | WMT17 | WMT17 | WMT17 | 637599 | 2003 | 2001 |
| En-Lt | WMT19 | WMT19 | WMT19 | 635146 | 2000 | 1000 |
| En-Ro | WMT16 | WMT16 | WMT16 | 610320 | 1999 | 1999 |
| En-Hi | WMT14 | WMT14 | WMT14 | 313748 | 520 | 2507 |
| En-Tr | WMT18 | WMT17 | WMT18 | 205756 | 3007 | 3000 |

Figure 5: Training data distribution over language pairs for OPUS-100 and WMT-14.

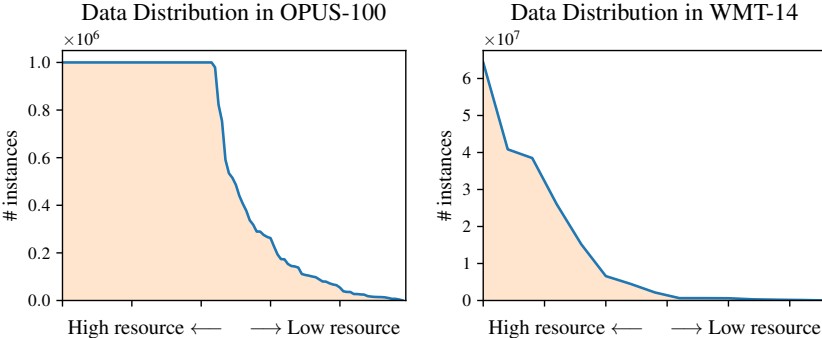

Table 3 lists the statistics for WMT-14. We collect these datasets following Siddhant et al. (2020). Figure 5 shows the training data distribution. Training data in WMT-14 is more imbalanced across different language pairs compared to OPUS-100.

# B    MORE RESULTS ON OPUS-100

Table 4: Selected sub-layers by Dedicated on OPUS-100. *SAN/CAN*: self-/cross-attention; *FFN*: feed-forward.

| Data Setting | Encoder | | | | Decoder | | | | | |
|---|---|---|---|---|---|---|---|---|---|---|
| | Bottom | | Top | | Bottom | | | Top | | |
| | SAN | FFN | SAN | FFN | SAN | CAN | FFN | SAN | CAN | FFN |
| Original, O2M | ✓ | ✓ | | ✓ | ✓ | | ✓ | ✓ | ✓ | ✓ |
| Original, M2O | ✓ | ✓ | | | | | | | ✓ | ✓ |
| Oversample, O2M | | | ✓ | ✓ | | | | ✓ | ✓ | ✓ |
| Oversample, M2O | ✓ | | | | | | | | ✓ | |

Figure 7 shows the translation performance of CLSR on each resource group when varying the budget $p$. We find that LS modeling greatly improves translation on low-resource settings, although

Figure 6: Heatmap of LSScore distribution on OPUS-100 trained w/ and w/o oversampling. X-axis denotes language pairs ranked by training data size, and y-axis denotes encoder (*enc*) and decoder (*dec*) sub-layers with a format of "enc/dec.layer types.layer index". Darker color indicates a larger LSScore.

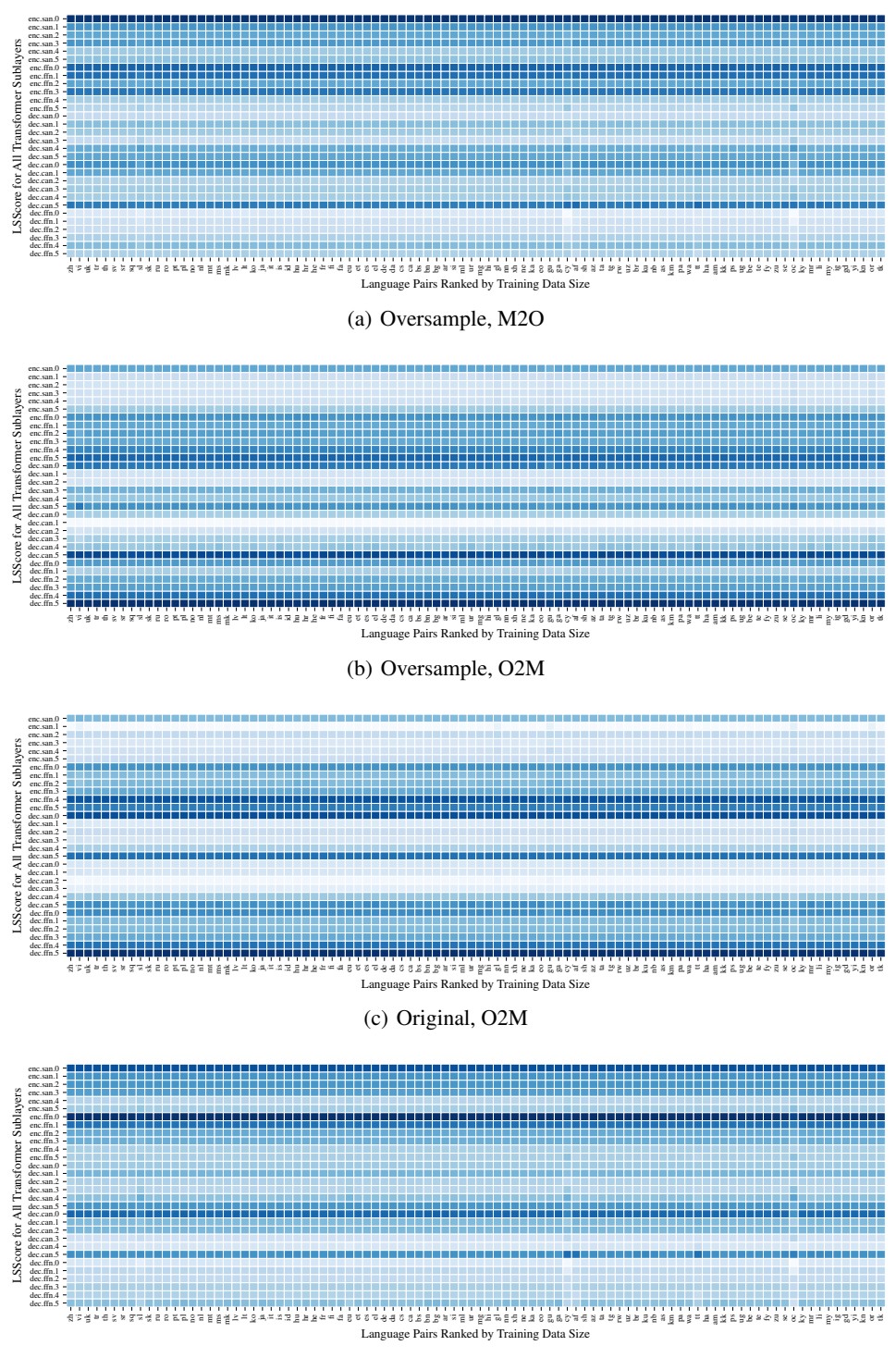

(a) Oversample, M2O

(b) Oversample, O2M

(c) Original, O2M

(d) Original, M2O

this improvement is partially offset by the oversampling strategy. We observe that the success of oversampling is built on top of sacrificing quality on high-resource languages, where CLSR largely narrows or even closes the performance gap against the multilingual baseline trained on the original

Figure 7: Impact of the budget $p$ on average test BLEU for High/Med/Low-resource languages on OPUS-100.

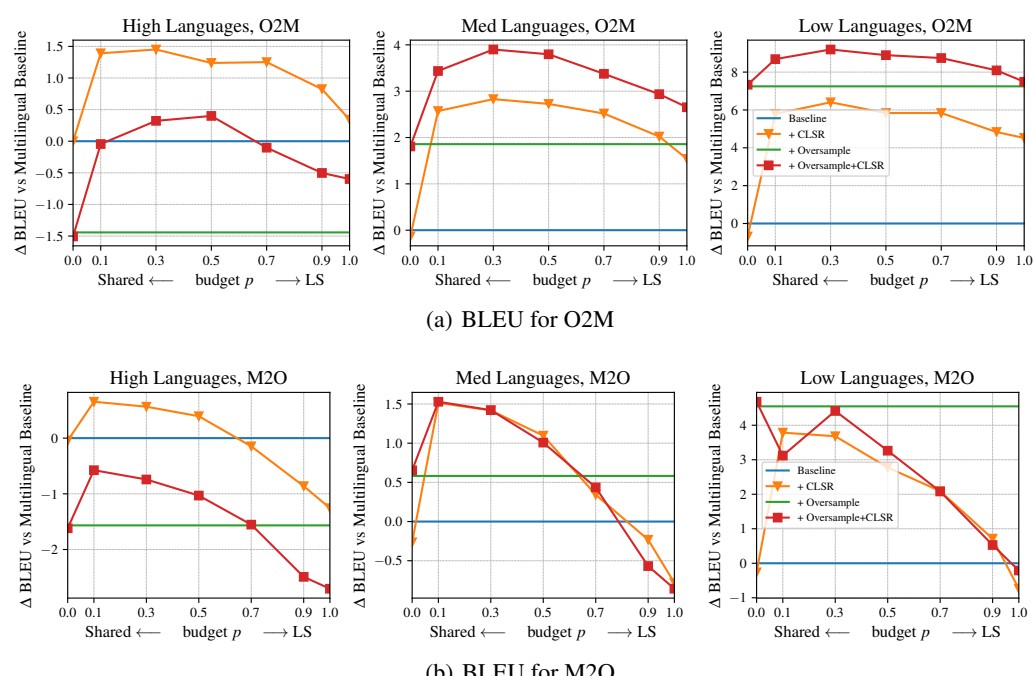

(a) BLEU for O2M

(b) BLEU for M2O

data. In addition, we find no significant difference in term of the trade-off curve across resource groups compared to Figure 2.

Figure 6 shows the heatmaps for LSScore distribution. Regardless of translation directions (O2M or M2O) and data settings (Original or Oversample), there is no clear linguistic pattern behind these heatmaps. The learning of CLSR is not linguistic driven.

## C  MORE RESULTS ON WMT-14

Table 5: Selected sub-layers by Dedicated on WMT-14.

| Data Setting | Encoder | | | | Decoder | | | | | |
| --- | --- | --- | --- | --- | --- | --- | --- | --- | --- | --- |
| | Bottom | | Top | | Bottom | | | Top | | |
| | SAN | FFN | SAN | FFN | SAN | CAN | FFN | SAN | CAN | FFN |
| Original, O2M | | ✓ | | ✓ | ✓ | | ✓ | ✓ | ✓ | ✓ |
| Original, M2O | ✓ | ✓ | | | ✓ | ✓ | | ✓ | | ✓ |
| Oversample, O2M | | ✓ | | ✓ | ✓ | ✓ | | ✓ | ✓ | ✓ |
| Oversample, M2O | ✓ | | | | | ✓ | | ✓ | | ✓ |

We show more results on WMT-14 here. Both heatmaps in Figure 9 and fine-grained trade-off curves in Figure 8 show a very similar story to the one on OPUS-100. In terms of LSScore distribution in Figure 10(d), we also have similar observation except that the cross-attention sub-layer in the decoder on M2O shows no clear LS preference.

These results suggest the high generalization of CLSR to different data settings.

Figure 8: Impact of the budget $p$ on average test BLEU for High/Med/Low-resource languages on WMT-14.

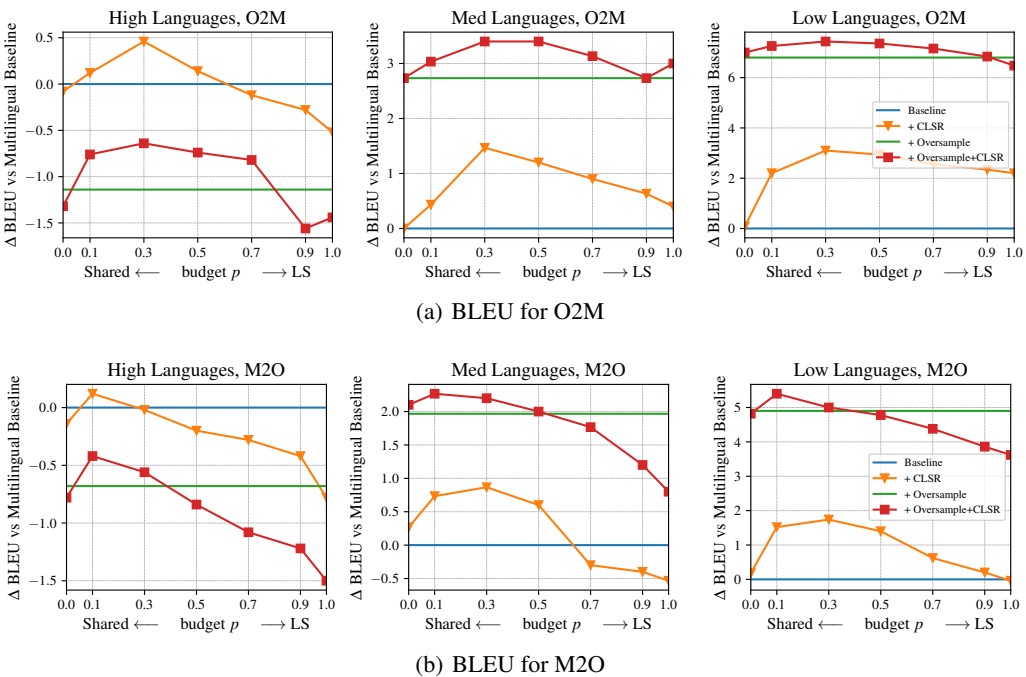

(a) BLEU for O2M

(b) BLEU for M2O

Figure 9: Heatmap of LSScore distribution on WMT-14.

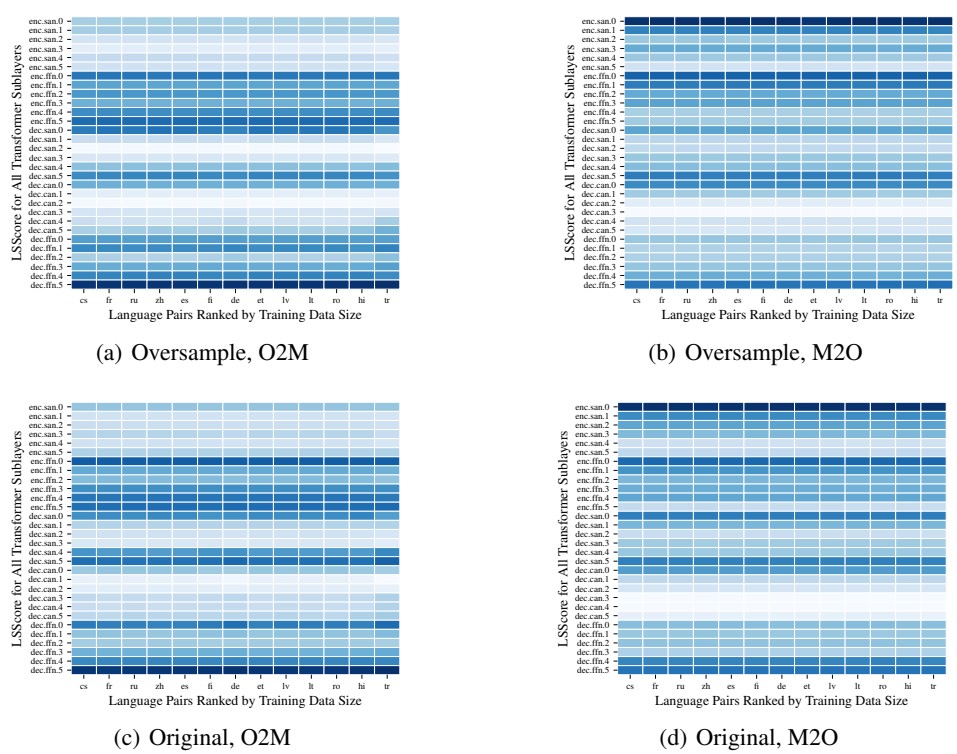

(a) Oversample, O2M

(b) Oversample, M2O

(c) Original, O2M

(d) Original, M2O

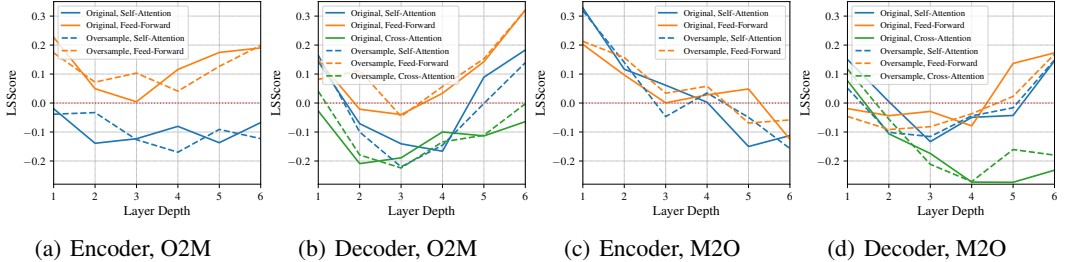

|   (a) Encoder, O2M | (b) Decoder, O2M | (c) Encoder, M2O | (d) Decoder, M2O |

Figure 10: LSScore of encoder and decoder sub-layers for O2M 10(a), 10(b) and M2O 10(c), 10(d) on WMT-14.

Table 6: Number of parameters for each model on OPUS-100 and WMT-14.

| Model | OPUS-100 | | WMT-14 | |
|---|---|---|---|---|
| | O2M | M2O | O2M | M2O |
| Baseline | 99M | | 106M | |
| LS$^\diamond$ | 129M | | 111M | |
| CLSR-S | 102M | | 109M | |
| CLSR-L | 148M | | 111M | |
| CLSR$^\star$ | 157M | | 121M | |
| Top-Bottom | 153M | | 116M | |
| Dedicated (Original) | 153M | 154M | 117M | 117M |
| Dedicated (Oversample) | 154M | 155M | 117M | 117M |

## D  MODEL PARAMETERS

We list model parameters in Table 6. Note that the source/target vocabulary for O2M is the same as the target/source vocabulary for M2O, thus models for O2M and M2O have the same number of trainable parameters except for Dedicated where the network structure differs.

## E  ABLATION STUDY

We provide two variants of CLSR for ablation. The first one examines whether CLSR could induce an optimal budget automatically by eschewing the budget constraint in Eq. 6 (CLSR w/o $p$); the second one further replaces the hard gating network with a vanilla soft gating network (CLSR-Gate).

Table 7 summarizes the translation results. Without the regularization term, CLSR w/o $p$ yields an average gating value of 0.37/0.16 and 0.39/0.19 for O2M/M2O on OPUS-100 and WMT-14, respectively[9], resonating with our finding that O2M requires more LS capacity compared to M2O. Compared to CLSR$^\star$, CLSR w/o $p$ produces more open gates, achieving comparable but overall worse performance in all data settings. This highlights the importance of the added regularization term, where without such regularization the model is likely spending extra learning cycles to automatically optimize for the budget as with CLSR w/o $p$. This in return hints at the importance of optimization routines (learning) which we held constant in this study.

By contrast, switching to soft gating network, CLSR-Gate gains translation quality in most settings, although the quality gains are mostly less than 0.1 compared to CLSR$^\star$. Note that, CLSR-Gate relies on continuous gates, indicating a much higher flexibility in balancing parameter sharing patterns. However, we also notice that CLSR-Gate performs worse on M2O, even fails to beat the vanilla Baseline (All) on the oversampled WMT-14. Unlike CLSR-Gate, CLSR$^\star$ uses hard binary gates which eases capacity analysis and achieves consistent improvements across data settings.

---

[9]We also average the gating value over the original and oversampled data settings.

| Data Setting | Model | #Param | O2M | | | | | M2O | | | | |
|---|---|---|---|---|---|---|---|---|---|---|---|---|
| | | | High | Med | Low | All | WR | High | Med | Low | All | WR |
| OPUS-100 | | | | | | | | | | | | |
| Original | Baseline | 99M | 21.39 | 22.36 | 18.02 | 20.93 | - | 28.55 | 30.10 | 29.71 | 29.27 | - |
| | CLSR* | 157M | +1.45 | +2.83 | **+6.40** | +2.97 | 95.74 | +0.65 | +1.52 | **+3.79** | **+1.61** | **96.81** |
| | CLSR w/o $p$ | 157M | +1.37 | +2.77 | +6.28 | +2.88 | 96.81 | +0.53 | +1.27 | +3.17 | +1.34 | 90.43 |
| | CLSR-Gate | 157M | **+1.55** | **+3.13** | +6.21 | **+3.06** | 97.87 | **+0.74** | **+1.62** | +2.87 | +1.48 | 93.62 |
| Over Sample | Baseline | 99M | 19.95 | 24.22 | 25.27 | 22.41 | - | 26.98 | 30.69 | **34.26** | 29.71 | - |
| | CLSR* | 157M | +1.76 | +2.04 | +1.94 | +1.89 | 93.62 | +0.82 | +0.84 | -0.13 | +0.62 | 76.60 |
| | CLSR w/o $p$ | 157M | +1.72 | +1.92 | +2.06 | +1.86 | **97.87** | +0.73 | +0.56 | -1.10 | +0.27 | 71.28 |
| | CLSR-Gate | 157M | **+2.00** | **+2.24** | **+2.35** | **+2.15** | **97.87** | **+1.12** | **+1.28** | -1.04 | **+0.68** | **79.79** |
| WMT-14 | | | | | | | | | | | | |
| Original | Baseline | 106M | 27.24 | 16.27 | 8.72 | 17.58 | - | 30.64 | 24.37 | 18.66 | 24.58 | - |
| | CLSR* | 121M | **+0.46** | **+1.46** | +3.10 | +1.71 | 100.0 | **+0.12** | **+0.73** | +1.52 | +0.80 | **84.62** |
| | CLSR w/o $p$ | 121M | +0.32 | +1.23 | +3.00 | +1.57 | 100.0 | -0.04 | +0.56 | +1.72 | +0.78 | 76.92 |
| | CLSR-Gate | 121M | **+0.46** | +1.43 | **+3.14** | **+1.72** | 100.0 | +0.02 | **+0.73** | +1.68 | **+0.83** | **84.62** |
| Over Sample | Baseline | 106M | 26.10 | 19.00 | 15.52 | 20.39 | - | 29.96 | 26.33 | 23.56 | 26.66 | - |
| | CLSR* | 121M | +0.50 | +0.67 | +0.64 | +0.59 | 100.0 | **+0.26** | **+0.30** | **+0.50** | **+0.36** | **100.0** |
| | CLSR w/o $p$ | 121M | +0.48 | +0.63 | +0.66 | +0.59 | 100.0 | +0.02 | +0.24 | -0.12 | +0.02 | 38.46 |
| | CLSR-Gate | 121M | **+0.52** | **+0.77** | **+0.78** | **+0.68** | 100.0 | +0.14 | **+0.30** | -0.64 | -0.12 | 61.54 |

Table 7: Translation quality for O2M and M2O on OPUS-100 and WMT-14 with the original and oversampled training data. *CLSR-Gate*: CLSR with $g(\mathbf{z}^l)$ a normal soft gating network; *CLSR w/o p*: CLSR trained without budget constraint.

