# OpenReview forum: "Share or Not? Learning to Schedule Language-Specific Capacity for Multilingual Translation"
_ICLR.cc/2021/Conference — ICLR 2021 Oral_

### Official Review · AnonReviewer1 · 2020-10-25
**A systematic analysis of language specific parameters in multilingual translation**

**Rating:** 8
**Confidence:** 5

**Review:**

In this paper, the authors present a study of different aspects of language-specific model capacity for massively multilingual machine translation. To this end, language-specific behaviour is achieved via a combination of conditional computation to decide whether to use language-specific parameters or not and statically assigning experts for each languages. The language specific sub-layers are incorporated throughout the network. The training objective allow budgetary constraints on the amount of language-specific parameters. The paper does a systematic analysis on the role of language specific parameters using the proposed architecture. Based on the analysis, recommendations on design of multilingual NMT architectures are proposed and their efficacy validated experimentally. The study sheds light on the amount of language specific parameter sharing, their distribution in the network, impact of language, etc.

I find the analysis presented in the paper very interesting and insightful - and distinguishes it from previous work in this area. The hypothesis are clearly stated and the experiments are well designed. The findings from the analysis are an important addition to the understanding of the role of language specific parameters in multilingual NMT.

In terms of modelling, the work follows in the line of recent work on language-specific parameters for multilingual NMT. The deviation from existing work is mixing elements of conditional computation with language specific computation. I see this work more as an analysis on language-specific parameters for a particular LS-model rather than a novel architecture. It is not clear how this model would compare to other models using language specific parameters (sparsely gated mixture of experts (Lepikhin et al 2020), light-weight adapters (Bapna et al 2019)  ).


Questions:

- Previous work has tried to combine both language-specific and shared parameters (Wang et al 2018), rather than making a binary choice between these. Did the authors compare with such an approach?
- Since a major part of the model contains shared parameters, was there a need for new set of shared parameters along with the language-specific parameters. The gating decision could have been to bypass the language-specific sublayer or not.

References
Yining Wang, Jiajun Zhang, Feifei Zhai, Jingfang Xu, and Chengqing Zong. Three strategies to improve one-to-many multilingual translation. EMNLP. 2018.

---

> ### Author Response · Authors · 2020-11-18
> **Response to Reviewer #1**
>
> Thanks for your insightful feedback.
>
> The goal of our study is to improve the understanding of the trade-off between LS and shared capacity for multilingual translation, and specifically answer the question of when and where LS capacity matters for multilingual NMT. We hope our study could shed light on developing novel multilingual architectures for future research.
>
> Please notice that the sparsely gated MOE model (Lepikhin et al., 2020) aims at scaling up Transformer, which has no language-specific components. The adapter (Bapna et al., 2019) is mainly used in a fine-tuning process on top of a pre-trained shared NMT model to largely benefit from the shared model parameters. It is separately fine-tuned for each language pair, and would significantly increase the training complexity in a massively multilingual setting.
>
> Regarding each of your questions:
> * Thanks for pointing out this missing related work (Wang et al., EMNLP 2018). We will extend our related work section to mention it soon. Using a fixed mix of language-specific and shared parameters (Wang et al 2018) is a valid strategy, but orthogonal to our research question to explore where language-specific computation is especially helpful.
> * Leaving the shared projection empty is also a potential choice, but that would introduce network structure differences between models using and not using the LS projections. Concretely, models routing through more LS paths will contain more linear sublayers than those skipped variants, and this difference might affect the training of the gating models and result in uncontrollable biases towards favoring or avoiding LS paths.

---

### Official Review · AnonReviewer2 · 2020-10-27
**Interesting contribution in the context of multilingual NMT**

**Rating:** 7
**Confidence:** 4

**Review:**

Manual parameter sharing schemes are generally costly to come up with and when they are obtained for certain language pairs they do not necessarily generalize well to arbitrary language pairs in multilingual NMT. The idea of learning which parameters to share across languages in multilingual transformer models is original and potentially useful for designing and analyzing multilingual models in the context of NMT.

**Strengths**:

The paper is well-written and easy to follow. The idea was (reasonably) well-positioned with respect to prior work and clearly presented.

The technical merit is essentially in coming up with the budget constraint term in the loss function that forces the multilingual encoder-decoder "super-network" to use the desired percentage of language-specific computation using gating.

A significant part of the contribution was in the analysis of the results, obtained by this learning-based parameter sharing approach, which was quite informative and revealed some interesting insights about where and when a language-specific computation is required. The takeaways should be of interest to researchers and practitioners interested in designing and analyzing multilingual NMT systems.

**Weaknesses**:

(1) Even though it is the first time such a method is applied in the context of NMT, the idea is not as much novel in the broader context of deep learning. Prior work has explored "learning-to-share"  strategies for parameter sharing in multi-task learning (see Ruder et al., AAAI 2018), and using gating/masking to control computational paths in a differentiable way (see Fan et al., ICLR 2019, Sukhbaatar et al., ACL 2019); it is clear that the focus is NMT but it should be worth mentioning/discussing such studies to better situate the work and to help the reader assess the actual contributions.

(2) Another weakness is that the comparison with the vanilla and LS baselines does not seem to be properly controlled in terms of parameters. I appreciate that the authors do not read too much into it and focus more on the analysis of the results, but one thing that remains unanswered in this paper is how the proposed method fairs against multilingual baselines that utilize (roughly) the same number of parameters; currently, the best models outperform the LS baseline by ~28M and ~10M parameters on OPUS-100 and WMT-14 respectively. How important is this difference?

(3) In the experiment about linguistic similarity, it appears that the capacity schedule is the same across languages and the authors conclude from this that the schedule has little to do with linguistic characteristics. However, the main driving force in the choice of the language-specific computation is currently a single hyper-parameter p which is the same across languages; so, this will lead to choices that are good on average for all language pairs involved for a given *universal* budget. Do you think the conclusion would be still the same if a language-specific hyper-parameter p_l was used instead?

---

> ### Author Response · Authors · 2020-11-18
> **Response to Reviewer #2**
>
> Thanks for your review and insights. Our response to each weakness you mentioned is below.
>
> (1) Thanks for pointing out these related studies in a broader context [1,2,3]. We didn’t find the mentioned work (Ruder et al., AAAI 2018) and (Fan et al., ICLR 2019). Instead, we found (Ruder et al, AAAI 2019, [1]) and (Fan et al., ICLR 2020, [2]). Please correct us if we misunderstood.
> We will include a discussion of these studies in our updated version soon based on your feedback.
>
> (2) The increase of model parameters doesn’t explain CLSR’s performance improvements. Please notice that, apart from LS$^\diamond$, we also offer CLSR-L for a more fair comparison. CLSR-L extends LS$^\diamond$ by applying the language-specific projection to each sub-layer of the Transformer. It utilizes roughly the same amount of parameters as CLSR* does. Our results show that CLSR-L yields inferior translation performance in almost all settings compared to CLSR*.
>
> (3) Making the hyperparameter p language-specific seems an interesting direction, although we don’t have much prior knowledge on how to adequately distinguish it among different language pairs. It might be possible that using a smaller p_l for low-resource languages could improve knowledge transfer towards Low languages for M2O translation.
> But based on our experiments, we would argue that the conclusion about linguistic correlation has a large chance to hold even if we adopt p_l rather than p. Under the budget constraint enforced by p as in Eq. (6), CLSR has enough freedom to schedule different amounts of LS capacity across different language pairs and different sublayers. However, our results show that the arrangement of LS capacity ends up being quite similar among different languages and sublayers, even though we vary p.
>
> [1] Ruder et al. Latent Multi-task Architecture Learning. In AAAI 2019
>
> [2] Fan et al. Reducing Transformer Depth on Demand with Structured Dropout. In ICLR 2020
>
> [3] Sukhbaatar et al. Adaptive Attention Span in Transformers. In ACL 2019

---

> > ### Comment · AnonReviewer2 · 2020-11-22
> > **Response to authors' feedback**
> >
> > Thanks for taking the time to answer my questions.
> >
> > (1) Yes, these are the references I was talking about.
> >
> > (2) The evidence presented in the paper is not sufficient to support that parameters do not play a role. The CLSR-L is only one out of the four baselines and it still has 9M and 10M fewer parameters than the CLSR* method for OPUS-100 and WMT14 datasets respectively. This is a somewhat stretched notion of "roughly the same amount of parameters". My recommendation for this issue would be to tone down the claims regarding improvements or simply compare against baselines that have a comparable number of parameters (e.g. within ~0.5-1M range).
> >
> > (3) Correct me if I am wrong but, apart from budget p, the gate parameters in Eq. 5 are also the same across languages and layers which creates an inductive bias towards gating z^l across languages in a unified way. It is also evident from all subfigures in Figure 6 that the gates function in (almost) the exact same way consistently across languages which seems like a hard constraint to achieve without enforcing it explicitly.  One more simple explanation of the observed behavior in Fig. 6 is that the parameters W_1, b, and w_2 do not provide enough flexibility for controlling the gate in different ways across languages and layers. Simply put, there is no language-specific behavior because there is no language-specific gating taking place. In other words, the capacity schedule has little to do with linguistic characteristics because it is designed to do so.

---

> > > ### Author Response · Authors · 2020-11-24
> > > **Response to Reviewer #2**
> > >
> > > Thanks for your constructive comments!
> > >
> > > (1) We updated our paper with a discussion of your mentioned studies in the related work.
> > >
> > > (2) Thanks for your suggestion! We toned down the performance improvement claim in the updated version.
> > >
> > > (3) Your understanding of the gate parameters is correct: these parameters are shared across language pairs. We agree that sharing these parameters might impose some inductive bias discouraging the language-specific behavior of CLSR. We added some additional discussion in the updated version to illustrate this point.

---

### Official Review · AnonReviewer4 · 2020-10-29
**Cross-language parameter-sharing for multi-lingual translation**

**Rating:** 9
**Confidence:** 3

**Review:**

The work proposes a hybrid architecture that has: (1) language-specific (LS) components; (2) as well as the components that are shared across all the languages -- a trade-off between specificity and generality.  A key conclusion of the work is that the best architectures typically are. the ones that have ~10-30% language-specific capacity.

In terms of experimental work, the work uses WMT-14 and OPUS-100 datasets to show the proposed trade-off.

In terms of exposition of the ideas, it's a well-written paper for the most part.

One issue that the authors could improve on is clarifying how "the amount of LS computation" is measured. You have mentioned it several times in the abstract/intro and it's neither clear nor referenced (it could be the number of parameters, it could be the number of basic computations, etc). For a new reader, it takes quite a while to find that $p$ is defined in eq. 6 and defined as a budget contains.

One other quibble is that all the trade-off figures are shown based BLEU/automatic metrics, which are known to be inaccurate. It would be nice to repeat one of the included evaluation with human judgments.

Overall, I view this as a good contribution to pave the way towards stronger, but reasonably-sized multilingual models. This is partially assuming that the authors will stay true to their promise that "Source code and models will be released."

---

> ### Author Response · Authors · 2020-11-18
> **Response to Reviewer #4**
>
> Thanks for your insightful feedback.
>
> * About the clarification on “the amount of LS computation”
>
>   Thanks for pointing this out. The term “the amount of LS computation” refers to the proportion of open gates where CLSR selects to route information through the LS path instead of its shared counterpart, which is directly regularized and guided by the budget constraint $p$. We will make this much clearer in our updated version soon.
>
> * About the human judgements
>
>   We agree that a human evaluation would be interesting, but want to point out that we evaluate each system on 188 (OPUS-100) and 26 (WMT-14) translation directions, respectively. Repeating this evaluation with human judgments would be a daunting task.
>
> * About the source code and models
>
>   Yes, we will release our source code and pretrained models to ease further study.

---

### Official Review · AnonReviewer3 · 2020-10-29
**Nice work showing how to add language-specific modeling capacity to large multilingual NMT models in a principled manner**

**Rating:** 7
**Confidence:** 4

**Review:**

In this work, the authors present a conditional language-specific routing (CLSR) scheme for transformer-based multilingual NMT systems. They introduce a CLSR layer after every transformer encoder and decoder layer; each such layer is made up of hard gating functions conditioned on token representations that will either select a language-specific projection layer or a shared projection layer. Further, a budget is imposed on the language-specific capacity measured by aggregating the number of gates that allow for language-specific computations; this budget constraint forces the network to identify the sub-layers that will benefit most from being language-specific.

This is nice work. The proposed technique has been described clearly, the idea is intuitive and the experiments are pretty compelling. I have a couple of minor comments/suggestions for the authors.

* The authors show heat-maps of LSScore distribution in Figure 6 (Appendix B) which suggest that the LS capacity schedule might have little to do with linguistic characteristics. However, this might have to do with the multilingual model being trained on as many as 94 different languages. It seems plausible that linguistic similarities might govern LS capacity scheduling when there are fewer training languages to learn from. To check for this, it might be interesting to redo this experiment with the medium resource and low resource buckets containing 26-28 languages each.

* There are two (among many other) interesting things that stand out from the results in Tables 1 and 2. (1) From Table 1, the only setting where CLSR* (as well as "Top-Bottom" and "Dedicated") underperforms compared to the baseline is M2O for low-resource languages. It seems like the use of language-specific layers here has a strong adverse effect on performance (-4.56 with CLSR-L) which is largely offset by CLSR*. Some more insights based on the individual BLEU scores for each test language in the "Low" bin and whether there were certain languages that were largely responsible for the drop in performance would be interesting to the reader. (2) From M2O in Table 2, the win ratios of Top-Bottom are much lower when compared with Dedicated and CLSR* (61.54 vs. 84.62 vs. 84.62; 30.77 vs. 84.62 vs. 100).  Could the authors share their thoughts on why this drop might be appearing?

---

> ### Author Response · Authors · 2020-11-18
> **Response to Reviewer #3**
>
> Thanks for your insightful feedback and constructive suggestions.
>
> For each of your concerns:
> * The consideration that the number of training languages might affect the scheduling behavior of CLSR totally makes sense. In fact, to make our claim convincing, we have already included one result with fewer training languages in our initial submission. Please notice that, apart from Figure 6 (Appendix B) on OPUS-100, we also took the same experiment on WMT-14 (14 languages involved) and showed the result in Figure 9 (Appendix C). The heatmaps in Figure 6 and Figure 9 reveal very similar patterns, both supporting that the LS capacity schedule in CLSR has little to do with linguistic characteristics.
> * Your observations on M2O translation are insightful. Our experiments show that LS capacity doesn’t work well in some M2O low-resource settings. Since different language pairs share the same target language in the M2O setting, we argue that sharing parameters could largely encourage positive knowledge transfer and deliver better translation quality towards low-resource languages whose training data might be too scarce to well-train their LS components.
> 1) Regarding M2O Low results in Table 1
>
>   Following your suggestion, we inspect the individual BLEU scores for M2O Low test languages. Below shows the result on the oversampled OPUS-100, where we list the results for the 8 languages to save space and report relative improvements against the baseline.
> |                | li   | my   | ig    | gd    | yi   | kn   | or   | tk   | WR on Low |
> |----------------|------|------|-------|-------|------|------|------|------|-----------|
> | LS$^\diamond$  | -6.2 | -4.  | -8.   | -15.6 | -9.4 | -5.3 | -2.3 | -4.5 | 4.76      |
> | CLSR-L         | 0.7  | -4.  | -10.6 | -22.8 | -3.1 | -2.7 | -2.7 | -9.  | 14.29     |
> | Top-Bottom     | -1.1 | -1.3 | -2.2  | 2.4   | -7.7 | -4.7 | -0.7 | -2.1 | 14.29     |
> | Dedicated      | -0.5 | -0.1 | -4.4  | 2.8   | -0.1 | -1.4 | -0.6 | -3.5 | 23.81     |
> | CLSR*          | 4.9  | -1.7 | -1.6  | 2.2   | 1.2  | -2.3 | -0.9 | -0.8 | 23.81     |
>
>   In line with your hypothesis, we observe that LS models suffer from large performance drop on some languages like gd, ig and tk, particularly with LS$^\diamond$ and CLSR-L. But we also notice that the win ratio on Low is small (<23.81%),  revealing the importance of sharing parameters for M2O Low.
>
> 2) Regarding M2O WR results in Table 2
>
>   Compared to OPUS-100, changes in BLEU tend to be more similar across languages on WMT-14, meaning that the same average improvement has a higher effect on the WR. Even in absolute terms, there is a clear gap between Top-Bottom and Dedicated/CLSR*. We argue that this is caused by the suboptimal schedule of LS capacity in Top-Bottom, which is partially supported by the evidence in Figure 10(c,d) and Table 5 (Appendix C). This is consistent with our finding that both the amount and the position of LS capacity matters for multilingual translation.

---

### Comment · Area_Chair1 · 2020-11-21
**The discussion stage is open!**

Dear Reviewers:

Thanks for your insightful reviews! Now the discussion stage is open and the authors have posted their responses. We will appreciate that the following things-to-do can be done by Tues, Nov 24.

1 Acknowledge explicitly that you have read the responses.

2 Modify your review if necessary.

3 Communicate with the authors/reviewers/AC by adding/responding to the comments if necessary.

Thanks a lot!

---

### Decision · Program_Chairs · 2021-01-07
**Final Decision**

**Decision:**

Accept (Oral)

**Comment:**

This paper proposes a conditional language-specific routing (CLSR)  mechanism for multilingual NMT, which also considers the trade-off between language specificity and generality.

All of the reviewers think this paper is interesting for both idea and empirical findings. Therefore, it is a clear acceptance.